# Biodegradable Polymer Electrospinning for Tendon Repairment

**DOI:** 10.3390/polym15061566

**Published:** 2023-03-21

**Authors:** Yiming Zhang, Yueguang Xue, Yan Ren, Xin Li, Ying Liu

**Affiliations:** 1Henan Institute of Advanced Technology, Zhengzhou University, Zhengzhou 450003, China; 2GBA National Institute for Nanotechnology Innovation, Guangzhou 510700, China; 3School of Biomedical Sciences and Engineering, South China University of Technology, Guangzhou International Campus, Guangzhou 511442, China; 4Zhejiang International Science and Technology Cooperation Base of Air Pollution and Health, Zhejiang Chinese Medical University, Hangzhou 310053, China

**Keywords:** electrospinning, tendon repair, nanofiber, 3D scaffolding, tendon scar healing

## Abstract

With the degradation after aging and the destruction of high-intensity exercise, the frequency of tendon injury is also increasing, which will lead to serious pain and disability. Due to the structural specificity of the tendon tissue, the traditional treatment of tendon injury repair has certain limitations. Biodegradable polymer electrospinning technology with good biocompatibility and degradability can effectively repair tendons, and its mechanical properties can be achieved by adjusting the fiber diameter and fiber spacing. Here, this review first briefly introduces the structure and function of the tendon and the repair process after injury. Then, different kinds of biodegradable natural polymers for tendon repair are summarized. Then, the advantages and disadvantages of three-dimensional (3D) electrospun products in tendon repair and regeneration are summarized, as well as the optimization of electrospun fiber scaffolds with different bioactive materials and the latest application in tendon regeneration engineering. Bioactive molecules can optimize the structure of these products and improve their repair performance. Importantly, we discuss the application of the 3D electrospinning scaffold’s superior structure in different stages of tendon repair. Meanwhile, the combination of other advanced technologies has greater potential in tendon repair. Finally, the relevant patents of biodegradable electrospun scaffolds for repairing damaged tendons, as well as their clinical applications, problems in current development, and future directions are summarized. In general, the use of biodegradable electrospun fibers for tendon repair is a promising and exciting research field, but further research is needed to fully understand its potential and optimize its application in tissue engineering.

## 1. Introduction

Under continuous economic pressure and a global aging environment, the treatment of musculoskeletal diseases is still a major challenge. As a component of the musculoskeletal system, the tendon is the connective tissue connecting muscle and bone, which can store and transfer energy during exercise and play a buffer role. As the tendon bears repeated load strength, it will lead to tearing and pathological changes of the tendon, and then cause pain, swelling, loss of tissue integrity, and functional damage [1,2]. At present, the traditional treatment methods for repairing or replacing damaged tendons, including autotransplantation, allograft transplantation, and prosthetic devices, rarely repair the shape and structure of normal tendons and have some inevitable shortcomings, such as re-rupture of the injured site, adhesion formation, scar formation, and long-term functional defects [3,4]. As tendon tissue repair involves complex coordination between the internal tendon core tissue and the surrounding external synovial tissue, it is necessary to find new and more effective treatment methods to cure tendon injury.

With the maturity of electrospinning technology and the development of technology, electrospinning has a variety of spinnable materials, simple devices, low preparation cost, controllable process, and high production efficiency. The nanofibers prepared using this technology have the advantages of a large specific surface area, high porosity, and adjustable diameter, which can mimic the natural extracellular matrix (ECM) in morphology, so it has great potential in tendon repair [5,6]. Electrospinning equipment mainly consists of a high-voltage power supply, injection pump, spinneret, and receiving plate [7]. In the process of spinning, the high-voltage power supply causes the surface of the droplets at the spinneret to produce a coulomb repulsion force opposite to the surface tension. When the coulomb repulsion force is greater than the surface tension, the droplets spray a thin stream at the conical apex. If the electric field voltage and polymer viscosity are appropriate, the jet stream is stretched and extended, and finally sprayed onto the collection device in a curved and unstable shape to form ultrafine nanofibers [8,9,10,11].

Different preparation methods and raw materials have an impact on the morphology and properties of the final fiber. By selecting the appropriate polymers and process parameters, it is easy to produce fiber structures with different morphologies from tens of nanometers to several microns in diameter, and at the same time, it can produce ECM nanofiber scaffolds with high mechanical strength and bionics [12]. In addition, many studies have also incorporated biomolecules, cells, nanoparticles or drugs into electrostatically spun nanofibre membranes, which can effectively improve their anti-adhesion and repair effects [13,14,15]. The electrospun fibers are structurally controllable, and the electrospun structures of different dimensions will have different effects on cells. A large number of previous studies have proved that, compared with two-dimensional (2D) scaffolds, three-dimensional (3D) nanofiber scaffolds are more conducive to cell morphology change, migration and cell-cell interaction [16]. Traditional 2D biomaterials (e.g., fiber sheets and bio-membranes) are usually stacked very tightly, resulting in a pore size that is too small, which limits cell infiltration and three-dimensional tissue regeneration [17]. The use of a biomimetic 3D nanofiber scaffold can overcome this obstacle and provides the necessary microenvironment for natural cell growth. The terrain and biomechanical clues provided by 3D-oriented nanofibers can guide the tendon differentiation of stem cells and maintain the phenotype of natural tendon cells [18].

The purpose of this review was to summarize the different structures of tendon repair product scaffolds prepared using electrospinning technology and their applications in the biomedical field (Figure 1). Biodegradable polymer electrospinning with different 3D structures is used for tendon repair. Bioactive molecules can optimize the structure of these products and improve their repair performance. The combination of other advanced technologies has greater potential in tendon repair. Biomaterials mixed with electrospinning are reviewed to discuss the influence of different biomaterials on the structure and properties of electrospinning. By discussing the advantages of an electrospun 3D structure in different stages of tendon repair, the current commercial products are listed, highlighting the current development progress. Finally, the opportunities, challenges, and new directions of electrospinning in future development are discussed. It provides a reference for the development of electrospinning technology in tendon repair. Of note, while electrospun fibers have the potential to be an effective tool for tendon repair, there are some potential limitations to consider, and further research is needed to fully understand its potential and optimize its application in tissue-engineering applications.

## 2. Repair Process after Tendon Damage

Natural tendon tissue has a layered fiber structure, which is composed of collagen arranged into fibers (type I collagen accounts for 95%) and cells with highly anisotropic mechanical properties. Tendon cells belong to the fibroblast family and are the most common cell type in tendon tissue. They are responsible for forming the structural framework of tendons and can effectively transmit force during muscle contraction [19]. Compared with other musculoskeletal tissues, the repair process is relatively slow due to its inherent connective tissue structure, poor vascular network, and low cell metabolic rate. The process of tendon healing is complex, involving internal and external mechanisms. The intrinsic mechanism is due to the regeneration of the cell group originating from the tendon parenchyma and/or tendon sheath. The external mechanism occurs because the external cells recruit from outside of the injured tendon to the injured site. The extrinsic mechanism involves the migration of cell populations from the blood and surrounding tissues (such as peritendinous tissue and synovial sheath) [20]. In addition, different repair processes will occur after tendon injury in different parts. For example, flexor tendon injury is caused by fibroblasts migrating from the tendon sheath to the tendon and starting to heal, while rotator cuff injury is caused by fibroblasts being exported from the tendon to the bone surface [21].

The first stage of tendon healing is the short cell phase of tendon cells and inflammatory cells invading the wound. After the injury, the initial inflammatory phase begins after a large number of cells die. Subsequently, inflammatory cells, such as neutrophils, monocytes, and macrophages, infiltrate the damaged area. The second stage is the proliferation stage of fibroblasts, which involves the migration of cells to the injured site and the proliferation and deposition of collagen fibers. Finally, in the remodeling stage, which lasts from several months to several years, tendon cells and collagen fibers are arranged along the longitudinal axis of the tendon in the direction of stress, the production of type I collagen increases, and the number of cells decreases [22]. Tendon contains fewer cells and blood vessels than other tissues. Tendon will not regenerate after injury, but will heal through excessive and disordered ECM deposition [23]. During the repair process, the synthetic level of type I collagen is significantly lower than that of healthy tendons. Instead, fibroblasts synthesize randomly oriented type III collagen, which is easy to cause fibrosis of tendon tissue, resulting in poor tissue quality, structural loss, and reduced mechanical strength [24]. In addition, the long healing process of injured tendons usually leads to the formation of fibrous vascular scar tissue, but the shape and mechanical and functional properties of repaired tendons are difficult to return to a normal state, and the risk of re-injury is increased. The adhesion site causes insufficient lubrication between the tendon and surrounding tissues, which can lead to friction and pain [25]. Therefore, it is more beneficial for tendon repair to develop therapies that can regulate different stages and/or different needs to promote regenerative healing.

## 3. Application of Biodegradable Polymers in Tendon Repair 

### 3.1. Application of Natural Biodegradable Polymers in Tendon Repair

Common biodegradable polymers used in tendon tissue engineering include natural biodegradable polymers and synthetic biodegradable polymers. Compared with synthetic polymers, tendon tissue engineering scaffolds prepared from natural polymers have unique advantages: better cell activity and adhesion, as well as a higher biocompatibility and degradation rate than other materials [26]. Common natural biodegradable polymers used for tendon tissue engineering include collagen, silk protein, gelatin, hyaluronic acid, etc. [27]. In tissue engineering, natural biodegradable polymers are used to prepare tissue scaffolds, which can be combined with growth factors [28] and stem cells [29], or used as drug carriers [30] to reconstruct the natural microenvironment of tendons [31] and help cell adhesion, growth, and differentiation [32]. The scaffold made of natural biodegradable polymer plays a temporary role in supporting the damaged tendon in the human body. During the synthesis and recovery of natural tissue, it can slowly degrade and produce non-toxic metabolites, which are absorbed by surrounding tissues [33]. Here, we briefly discuss the role of some natural biodegradable polymers in tendon repair.

In tendon tissue engineering, collagen, as the main component of the human tendon, is a natural degradable polymer with a biomimetic tendon ECM structure. Due to its good cell and growth factor binding sites on the surface, it has the function of supporting cell adhesion, migration, growth, and differentiation [34]. Tissue engineering technology has been used to combine a collagen scaffold with growth factors to help repair damaged tendons. Sun et al. [35] used a functional collagen scaffold loaded with recombinant stromal cell-derived factor-1α containing a collagen-binding structural domain (CBD) to recruit mesenchymal stem cells (MSCs) and Achilles tendon fibroblasts, and promote in situ tendon regeneration. In addition, due to its biodegradable properties and good biocompatibility, it achieves both a sustained drug-retentive treatment of the damaged tendon area, promotes the formation of type I collagen, and reduces the inflammatory response due to the implanted scaffold.

Silk proteins have good strength and toughness compared to most natural biodegradable polymers [36] and have a variety of applications in tissue engineering, wound healing, and drug delivery [36]. Silk proteins mimicking the native interface may be more effective in supporting tissue regeneration, maintaining good tensile strength, and initiating tendon regeneration for at least 1 year in the in vivo environment of damaged tendons [29]. Teuschl et al. [37] used silk protein woven into a wire rope shape as an anterior cruciate ligament substitute and combined silk protein fibrous scaffolds with adipose-derived stem cells to promote ligament regeneration in goats over a period of 6 to 12 months, and progressive degradation of silk fibers occurred while regenerating tissue increased.

Gelatin is a unique fibrin, a collagen derivative that mimics the natural connective tissue microenvironment [38]. It significantly improves the infiltration, adhesion, spread, and proliferation of cells on tissue-engineered scaffolds and has low immunogenicity and biodegradability [39]. Liu et al. [27] prepared membranes made from gelatin using the electrostatic spinning technique with micron-sized 3D structures, which significantly promoted the regeneration of type I collagen in a rat chronic rotator cuff tear model. The gelatin scaffolds prepared using electrostatic spinning showed good cytocompatibility and were able to mimic the natural tendon tissue microenvironment to accelerate tendon healing and repair after damage. The gelatin scaffold’s similar structure to the tendon extracellular matrix aided endothelial cell vascularization and transport of active substances in the damaged and healing tendon. The degradation products of gelatin have no side effects on tendon tissue and are slowly enzymatically cleaved as the tendon is stimulated by the gelatin scaffold to return to normal function.

### 3.2. Limitations and Future Progress Related to the Use of Biodegradable Polymers in Tendon Repair 

Although biodegradable polymers have made great strides in the field of regenerative medicine, it is still difficult for any single polymer to meet all the necessary biochemical needs [40]. In order to develop high-performance stents that meet clinical needs, a number of key points need to be considered. Firstly, a rational design of the stent source material is required [41]. Secondly, overcoming problems with mechanical properties and degradation rates requires improved stent-preparation techniques [42]. Natural biodegradable polymers, such as collagen and gelatin, usually share the common disadvantage of lacking mechanical properties and having uncontrolled degradation rates [43]. Collagen, the first naturally occurring biodegradable polymer to be used as a scaffold, is not mechanically strong enough to provide post-operative mechanical support after a tendon tear [34]. To overcome this disadvantage, electrostatic spinning scaffolds have been extensively researched and applied. Electrostatic spinning scaffolds is a technique for spinning polymeric materials into a fibrous form through the means of an electrical charge, which is highly controllable and adjustable. By adjusting the spinning parameters, such as the charge density, flow rate, and distance, fibrous scaffolds of different diameters, morphologies, and porosities can be obtained [44]. The degradation rate and tissue repair of the electrostatic-spun scaffold can also be adjusted by controlling the spinning parameters [45]. For example, Xie et al. [46] designed a collagen/polylactic acid (Col/PLA) hybrid yarn that blended a natural polymer with a synthetic polymer to take advantage of the mechanical strength of the PLA and the biological activity of collagen. The hybrid tendon scaffold combining the biological and synthetic components had the sufficient tensile strength to maintain normal tendon function while promoting host tissue healing. With this in mind, we focus on the prospects for the application of electrostatic spinning technology in tendon repair in the following section.

## 4. Functional Electrospinning Products of 3D Structures for Tendon Repair

Electrospinning technology occurs when a viscoelastic polymer fluid is extruded from a spinneret under the electrostatic repulsive force of a strong electric field, and the surface tension produces a hanging droplet. After electrification, the electrostatic repulsion between surface charges with the same symbol causes the droplet to be deformed into a Taylor cone, and the charged jet is ejected from the Taylor cone [47]. Different preparation processes will produce multiscale structure repair products. In the application of tendon repair, the main products are nanoyarn, nanotube, spinning film, and a layered multiscale structure electrospinning scaffold (Figure 2A). Different structures have shown an excellent performance in tendon repair.

### 4.1. Nanofiber Yarn

Tendon is a highly anisotropic tissue, in which the collagen fibers are arranged uniaxially and parallel to each other. Tendon fibroblasts showed highly organized and randomly oriented morphologies in aligned and random parts, respectively [48]. Uniaxially aligned nanofibers can provide contact guidance and alignment for tendon fibroblasts, and can be used for tendon regeneration when their ECMs are organized into highly ordered structural arrays. Tendon fibroblasts showed highly organized and randomly oriented morphologies in aligned and random parts, respectively. Uniaxially aligned nanofibers can provide contact guidance and alignment for tendon fibroblasts, and organize their ECMs into highly ordered structures [49]. Porous engineering electrospun nanofiber materials have highly customized pore traps, topological porous channels, and a high surface area. Compared with oriented electrospun nanofibers, porous nanofibers generally show stronger activity and performance. Macroporous fiber materials usually show a higher loading capacity and stronger accessibility to active sites [50,51]. In addition, compared with linear uniaxially aligned nanofibers, the uniaxially aligned nanofibers with controllable curvature can better simulate the anatomical structure of collagen fibrils in tendon tissue, and better protect tendon fibroblasts under uniaxial strain [52], providing more cell-adhesion sites [53].

Inspired by the multiscale helical fiber structure of natural biological tissue, the helical nanoyarn with a multiscale structure can be prepared using electrospinning and continuous twisting technology. The structure has excellent mechanical properties and ultra-high extensibility. The multiscale periodic topological structure on the material surface can not only change the physical characteristics of cells (survival rate, volume, orientation, growth and shedding), but can also induce the directional differentiation of mesenchymal stem cells into muscle cells by regulating the transport of cell types and specific transcription factors to the nucleus [54]. Electrospun nanofibers coated with polycaprolactone (PCL) or poly (3-hydroxybutyric acid) (P3HB) and twisted silk fibroin (SF) yarn form a nano/micro hybrid structure. PCL or P3HB with good biocompatibility can provide the mechanical strength for the scaffold. A SF yarn contains the unique crimp structure of tendon tissue. The scaffold can promote the adhesion and attachment of fibroblasts and proliferate at a higher speed [55].

When tendons are injured, it is difficult to balance the relationship between exercise and rest. Fortunately, Wang et al. achieved the goal of repairing tissues while exercising properly. They first prepared a kind of electrospun nanofiber composite membrane with a PCL/polyurethane (PU)/PCL sandwich structure, then twisted the fiber membrane into a straight yarn, and then further twisted it to form a spiral yarn support. Due to the non-planar movement and high extensibility of the scaffold, when the rat bone marrow cells cover the whole scaffold to stretch the scaffold, the cells can maintain a high survival rate under the cyclic strain, and gradually proliferate with the passage of time (Figure 2B) [19]. In addition to the twisting technology, the rotating spindle collecting device can also make the aligned fibers curl. The resulting aligned crimped fibers have the same amplitude and wavelength as natural collagen fibers. The crimped fibers can be used to guide the proliferation of fibroblasts in vitro and the synthesis of ECM [56]. The combination of the hot drawing process and electrospinning can enhance the mechanical strength and biological properties of a nanoyarn [57,58].

### 4.2. Electrospinning Tube

In the surgical treatment of a ruptured tendon, the development of a nanofibrous tube that prevents penetration between the tendon and the surrounding tissues is a possible solution to improve the adhesions that may occur at the injury site in order to cope with the bleeding and infection during the healing process that lead to adhesions. Copolymers of polyhydroxybutyrate and ε-caprolactone (DegraPol^®^, DP) are biocompatible and biodegradable, and have proven to be ideal tendon materials for electrostatic spinning tubes [59,60]. By using electrostatically spun DP tubing to treat a rabbit model of Achilles tendon rupture, Buschmann et al. found that placing fibrous tubing around a transected Achilles tendon for 12 weeks did not result in overexpression of the inflammatory-associated protein, significantly reduced the formation of adhesions, and underwent partial degradation [61]. In another study that used the strategy of free delivery of growth factors or drugs based on the scaffold, the block copolymer DP was mixed with platelet-derived growth factor BB (PDGF-BB), and a kind of PDGF-BB bilayer bioactive electrospinning tube was developed (Figure 2C). The double-layer structure had a biologically active inner layer and a non-biologically active outer layer, which could continuously deliver PDGF-BB at the tendon rupture and injury site to promote collagen synthesis and increase cell proliferation, while providing a physical barrier for surrounding tissues, thus, significantly reducing peritendinous adhesion [62]. In addition, tubular repair structures were developed in combination with tubular braids in order to enhance the mechanical properties and repair ability of the tubular structures. The electrostatic spinning of a new acrylate capped polyurethane based polymer was able to represent an optimal and controlled environment. Mechanical testing of isolated sheep tendons has shown that the minimum limiting stresses required for flexor tendon repair were fully met. The incorporation of anti-inflammatory and antiadhesive compounds has a minimal inflammatory response and adhesion to surrounding tissues to minimize scar tissue formation and improve the rate of collagenization [63]. An electrospinning tube, as a kind of product to wrap the injured part of tendon, has the application potential of preventing peritendinous adhesion and repairing.

### 4.3. Electrospinning Nanofiber Membrane

Compared with random fibers, oriented fibers have better mechanical properties, faster charge transfer, and a more regular spatial structure. The oriented nanofiber structure will change the cell phenotype, thus, affecting the differentiation of stem cells [64]. A hydroxyapatite (HA)/polylactic acid (PLLA)-oriented nanofiber membrane prepared using electrospinning has a rough surface and weak hydrophobicity. The fiber membrane has good cell compatibility, and the cells grow along the fiber arrangement direction, which can effectively promote tendon–bone healing and reduce the rate of rotator cuff repair and tear [65]. When poly-D-L-lactide-glycolide (PLGA) is used as the polymer material of the nanofiber membrane, doxycycline drug elutes the nanofiber membrane. Histological examinations and monitoring of postoperative animal activity levels have shown that the nanofiber membrane can promote the healing of tendon rupture, and can be used as an auxiliary tool for Achilles tendon homotransplantation and reconstruction surgery [66].

Electrospun nanofiber membranes also play an antiadhesive role during tendon repair. However, most electrospun membranes promote cell adhesion due to their relatively hydrophobic surface [67]. The zwitterionic polymer chain is generated in situ from the photopolymerization initiated by the sub-surface of the nanofiber from the inside out, forming a solid network interpenetrating with the polymer chain of the nanofiber matrix, and then the super-lubricated nanofibrous membranes (SLNMs) is grown in situ on each electrospun nanofiber. Research using a rat tendon adhesion model and an abdominal adhesion model showed that a nanofiber membrane with SLNMs can prevent postoperative adhesion. Compared with two antiadhesion products (Interseed^®^ and DK^®^ membrane) used in the clinic, SLNMs can significantly reduce the adhesion around the tendon (Figure 2D). It is not only more effective, but also has the advantage of lower production costs [68]. In addition, a PCL/poly (2-methacryloxyethyl phosphate choline) (PMPC) composite nanofiber membrane with interconnected 3D porous structure, due to the presence of amphoteric ion PMPC, produces a hydrated lubricating surface. When the weight ratio of PMPC to PCL is 0.1, the minimum friction coefficient of the composite nanofiber membrane can reach 0.12. With the increase of mass ratio, the minimum friction coefficient can be less than 0.05. Highly lubricated surfaces effectively inhibit the adhesion of fibroblasts and significantly reduce tissue adhesion during tendon repair in vivo [69].

Compared with the single-layer nanofiber membrane, the double-layer membrane with random and arranged nanofibers in the outer and inner layers provides isotropic mechanical properties, which are more robust and tear resistant after being implanted in the body [70]. The change of pore size distribution in different layers is conducive to cell stratification [71]. Li et al. constructed a PLLA/nHA-PLLA bilayer organic/inorganic flexible bipolar fiber membrane (BFM) using electrospinning. Compared with PLLA’s simple fiber membrane (SFM), BFM significantly increased the glycosaminoglycan staining area at the tendon–bone interface, improved collagen tissue, and promoted the regeneration of attachment points in a rotator cuff tear [72].

### 4.4. Layered Electrospinning Scaffold

The multiscale structure and polymer type have different effects on the function and mechanical properties of tendon cells. Layered multiscale electrospinning scaffolds can not only highly simulate the hierarchical structure of ECM, but also induce different fibroblast morphological modifications under static and dynamic conditions, thus, changing their shape in the direction of the load application [73]. The layered structure porous scaffold is ingeniously designed and developed. Nanofibers can imitate cell tissue by bonding the biomimetic cell wall of micropores of different sizes (100–1000nm) [74]. In addition, a continuously graded calcium phosphate coating is formed on the non-woven mat of PLGA nanofibers. The mechanical properties and biological activities of the scaffold are significantly affected by the linear gradient of calcium phosphate content [75].

A multilayer nanofiber scaffold is usually made of electrospun high-conductivity solution. The polymer solution mixed with salt or bioactive molecules can adjust the fiber structure and function [76]. One multiscale fiber scaffold was composed of orderly arranged PCL microfibers/collagen bFGF nanofibers (mPCL-nCol-bFGF), and coated with sodium alginate to inhibit the surrounding adhesion. Rabbit tendon cells showed the highest expression of tendinous markers and cell proliferation under dynamic stimulation in vitro. Achilles tendon tissue was regenerated 12 weeks after operation, and collagen was arranged neatly (Figure 2E) [77]. Arranged PCL/Col multiscale fibers can mimic the tendon matrix and enhance porosity by sacrificing collagen nano fibers. Plasma treatment with argon and nitrogen enhances the roughness and hydrophilicity of the fibers, which can significantly enhance cell adhesion and proliferation, and induce tendon differentiation of MSCs [78]. Additionally, Liu et al., after freezing and vacuum drying a fresh amniotic membrane, effectively removed most of the cell components in the amniotic membrane to retain bioactive molecules, and built a growth factor release system that conformed to the tendon healing cycle. Combined with PCL nanofiber membrane, a biomimetic composite membrane with a tendon sheath and putamen structure was constructed, which enhanced the tensile resistance of the amniotic membrane. The 3D porous structure provided sufficient 3D space structure for cell growth. This upregulated the phosphorylation of ERK1/2 and SMAD2/3, promoted the adhesion and proliferation of tendon cells and fibroblasts, and improved collagen synthesis. The rabbit tendon repair model effectively isolated the exogenous tissue adhesion and promoted the endogenous healing of the tendon [79]. The significant advantage of layered and multiscale structures for electrospinning scaffolds is that each layer can have a different function and the different layers can work synergistically with each other to enable a wider range of applications.

The arrangement and topographic characteristics of the multilayer nanofiber scaffold can regulate its mechanical and biological activities [80,81,82]. A study confirmed that the multilayer arrangement scaffold is more conducive to cell proliferation and infiltration, enhanced collagen arrangement, and tendon-related gene expression than the multilayer non-arranged scaffold [80]. In another study, PCL or PLLA nanofiber sheets were stacked and woven together to form a multilayer scaffold aligned with electrospun nanofibers. They had similar nanoscale clues, but different macrostructures. The woven support showed higher tensile strength and suture retention strength, but lower modulus. Due to its larger surface area, the stacked scaffold was better in cell proliferation and migration, and promoted an increase in the total collagen content and total sulfated glycosaminoglycan content. There was no difference between the expression of tendon-related genes and the deposition of extracellular matrix protein [81]. To sum up, it is more beneficial to enhance the effectiveness of tendon repair by comprehensively considering the micro and macro structure of the multiscale scaffold for tendon tissue engineering.

**Figure 2 polymers-15-01566-f002:**
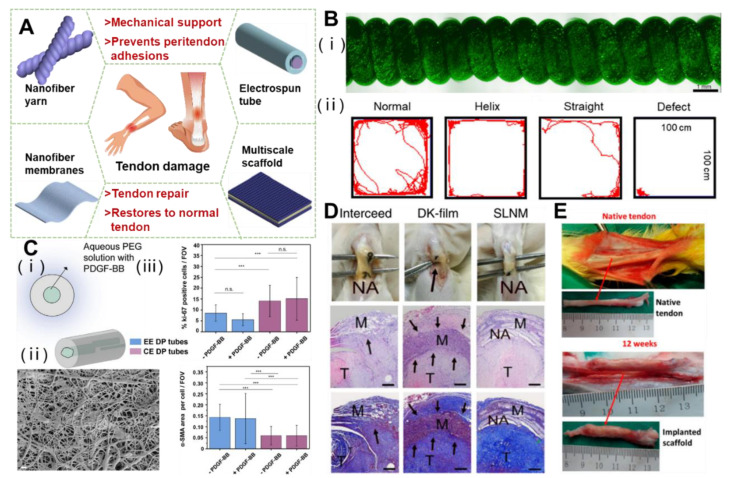
Functional electrospinning products of 3D structures and the effect on tendon repair. (**A**) Schematic diagram of electrospinning for various tendon repair. (**B**) The helical scaffolds are biocompatible and restore motor function in the rat Achilles tendon.Reprinted with permission from Ref. [19]. Copyright 2022 Elsevier Inc. (**C**) Characterization of coaxial electrospun bilayer tubes, which promote recruitment and proliferation of tendon cells. (*** *p* < 0.001). Reprinted with permission from Ref. [62]. Copyright 2020 Elsevier Inc. (**D**) In vivo antitissue adhesion properties of the superlubricated electrospun nanofibrous membranes. The black arrows point to adhesion site. Reprinted with permission from Ref. [68]. Copyright 2022 Springer Nature Limited. (**E**) The mPCL-nCol-bFGF implantation promoted tendon regeneration after 12 weeks. Reprinted with permission from Ref. [77]. Copyright 2019 American Chemical Society.

## 5. Electrospinning Composite Scaffold for Tendon Repair

Researchers have recently combined electrospun nanofibers with emerging materials (such as minerals, metals, growth factors, stem cells, drugs, and nanoparticles) to further enhance their physical and chemical properties and biological activities (Figure 3A). The structure and porosity of the scaffold should promote cell activity and the formation of new tissues.

### 5.1. Parameter Control and Temporary Storage Disadvantages of Electrospinning Materials

Due to the uncontrollable parameters during the preparation of traditional electrospinning technology, the prepared electrospinning products usually have small pore sizes and random fiber diameters, which is not enough for cell infiltration and growth [83]. Therefore, it is still a challenge to manufacture electrospun products with a porosity that can simulate the complex spatial distribution of natural tissue. In order to improve the application performance of electrospinning products in tissue engineering, a variety of improvement strategies have been proposed. Near-field electrospinning (precision electrospinning or direct-writing electrospinning) has proved to be able to accurately control fiber deposition to form patterns. By fixing the spinneret on the motion table, the collector/spinneret can run according to the pre-programmed track [84]. This technology is crucial for the application of classical spinning products in biomedicine.

The shape and diameter of products prepared using electrospinning technology mostly depend on the process parameters. With the update in technology, the fiber diameter, diameter distribution, and orientation of electrospinning products are within the controllable and adjustable range as a result of controlling the conductivity, electric field strength, solution concentration, auxiliary electric field, and other factors [85,86,87,88]. With the development of electrospinning technology, the process parameters that affect the final properties of electrospinning products are more and more accurately controlled, and products with stable physical and chemical properties can be obtained. The electrospinning products prepared can be produced and sold in batches.

However, the weak mechanical properties of electrospun materials are a problem that cannot be ignored, and they are usually not suitable for hard tissue regeneration. It is a feasible method to enhance the tensile strength and other mechanical properties of electrospun products through subsequent extrusion, heating, and other methods [46]. At the same time, the degradation rate of electrospun nano-supports in vivo is not controllable, which can be overcome by mixing other bioactive substances to synthesize new electrospun materials or developing new spinning technologies.

### 5.2. Scaffold Mixed with Biological Factors

The use of exogenous growth factors has the potential to accelerate cell proliferation, collagen synthesis, and extracellular matrix synthesis, thereby accelerating recovery and enhancing repair. The basic fibroblast growth factor (bFGF) is usually selected as a representative growth factor that affects and enhances tendon/ligament repair and healing, cell differentiation, diffusion, proliferation, and ECM production [89]. Considering the advantages of the above growth factors, Liu et al. doped bFGF with dextran glassy nanoparticles and an electrospun PLLA-glucan composite fiber. Loaded growth factors maintain their biological activity for 30 days, release approximately 48.71 ± 13.53%, and enhance the viability of fibroblasts, collagenase production, capillary endothelial cell proliferation, and tissue regeneration [90]. However, due to the discontinuity, inhomogeneity, and defects introduced into the fiber structure, the mechanical properties of the polymer fiber may be reduced by adding particles into it. In another study, Zhao et al. tested a bFGF-loaded PLGA electrospun fiber membrane with a core–shell structure. Studies have shown that a bFGF-PLGA scaffold can enhance the formation of collagen tissue and fibrocartilage. In addition, compared with the PLGA scaffold, the system shows higher mechanical performance and stiffness. The results showed that the biological activity of bFGF could be preserved for 21 days, significantly accelerating tissue healing and remodeling [91]. Taking advantage bFGF’s ability to enhance the toughness of the product, Zhou et al. uniformly adhered bFGF/vascular endothelial growth factor A (VEGFA) to the surface of the suture through polydopamine. At multiple time points of the chicken model of flexor tendon injury, the final strength of the tendon treated with the bFGF/VEGFA release suture was significantly greater than that of the control suture group (Figure 3B), the tendon slip offset was significantly longer than that of the control naked suture, and the finger flexion work was significantly reduced. In the sixth week, the adhesion score of the bFGF/VEGFA-releasing suture group was significantly lower than that of the control suture group. bFGF/VEGFA has the same effect when applied to the rat Achilles tendon injury model [92]. Extracellular signal-regulated kinase 2 (ERK2) can block tendon adhesion by inhibiting fibroblast proliferation and indirectly reducing the deposition of type III collagen (Col III). The incorporation of ERK2-small interfering RNA (siRNA)/cationic 2,6-pyridyldicarboxylic aldehyde-polyvinimide (PDA) into a PLLA electrospun film can protect the biological activity of ERK2-siRNA and continuously release siRNA to achieve a long-term adhesion prevention effect (Figure 3C) [93].

### 5.3. Electrospinning Scaffold with Cells

Due to the limited number of stem cells in tendons, the low activity/repair ability severely limits the ability of tendon regeneration. MSCs have a strong self-renewal ability, differentiate into various cell lines, and produce abundant functional paracrine factors. Therefore, MSCs-mediated cell therapy has great potential as a new strategy for tendon tissue regeneration. An SF/methacrylic acid hydrogel (GelMA) bioactive nanofiber scaffold (Figure 3D) designed by Xue et al., silk fibroin, provides support strength and ductility for the scaffold, while methacrylic acid hydrogel promotes cell attachment and growth. The composite scaffold composed of the two is easy to adhere without additional fixation, and can be biodegradable under control. Inoculating mesenchymal stem cells on it greatly improves the efficiency of tendon regeneration [36]. 

Adipose-derived stem cells (ADSCs) are easy to obtain, have a large storage capacity, and have the potential to differentiate tendon cells. The mesh scaffold composed of a longitudinally arranged PGA fiber inner layer and a braided PGA and PLA fiber outer layer is evenly inoculated with ADSC on the fiber scaffold, and then implanted into the injured Achilles tendon after expansion in vitro. For the treatment in the rabbit Achilles tendon injury model, with the increase of implantation time, the cell seed constructs gradually formed new tendons and became more mature at 45 weeks. A histological examination confirmed the similarity between the new tendon and the natural tissue. Compared with the normal tendon, the tensile strength of the treatment group reached 60% [94]. 

Skeletal muscle-derived cells (MDCs) are a mixed cell group containing muscle cells, fibroblasts, satellite cells, and muscle-derived mesenchymal stem cells. Compared with mature and fully differentiated tendon cells, MDCs have higher proliferation potential [95]. MDCs from mice were inoculated onto PGA electrospun fibers and sutured to the defect. Compared with tendon cells, they can promote the formation of new tendons and produce stronger tendon tissue. Compared with tendon tissue, thicker collagen fibers show higher maturity. After 12 weeks, MDCs lost the expression of myoD and desmin, and the expression of tendon specific markers increased [96]. 

### 5.4. Mixed Scaffold Mixed with Nanoparticles

The performance and function of the original polymer structure can be enhanced by adding nanoparticles into the electrospun fiber. Rinoldi et al. incorporated silica nanoparticles into the beaded electrospun fiber structure, which could improve its wettability, stiffness, and degradation rate, and was conducive to cell proliferation and extracellular matrix deposition [97]. Hydroxyapatite has rich active sites and ion-exchange properties. Using the sol–gel method, HAp nanoparticles were incorporated into PCL/CS nanofiber scaffolds to promote the formation of porous structures in the PCL/CS matrix. A physical and chemical characterization and its biological properties showed that the structure was similar to that of human normal ligaments and tendons, which was conducive to the tendon regeneration reaction of human osteoblasts [98].

Infection after tendon surgery seriously prolongs the repair cycle. Due to the significant antibacterial activity of silver nanoparticles (Ag NPs), many studies have added them to the nanofiber scaffold to effectively reduce the inflammatory reaction and, thus, reduce the adhesion of tissues around the tendon [99,100,101]. Microsol electrospinning can be incorporated to control the release of hydrophilic drugs in hydrophobic polyester fibers. Hydrophilic mitomycin C (MMC) was loaded into HA, then encapsulated in PLLA fiber using electrospinning. The composite fiber membrane can control the stable release of MMC, and a low dose of MMC can inhibit the adhesion, proliferation, and apoptosis of fibroblasts, which is conducive to tendon healing [102]. In addition, zinc oxide nanoparticles (ZnO NPs) inhibit bacterial activity by releasing zinc ions, and participate in various biological processes such as metabolism, cell proliferation and differentiation, and gene expression control. A PCL/HA/ZnO film was constructed using the electrospinning method. After primary mouse fibrochondrocytes and primary mouse tendon cells were cultured on the film, the PCL-5% HA-1% ZnO film (the weight ratio of HA and ZnO was 5% and 1% respectively) showed good antibacterial specificity and excellent cell compatibility and cell adhesion. The PCL-5% HA-1% ZnO film also promoted osteogenesis, cartilage formation, fibrocartilage formation, and tendon healing [103].

### 5.5. Electrospun Nanofiber/Hydrogel Composite 3D Scaffold

In order to improve the biological reaction of electrospun nanofiber structure, the use of hydrogels with excellent performance is also a feasible strategy. Rinoldi et al. coated a thin layer of GelMA hydrogel containing mesenchymal stem cells on the electrospun nanofiber substrate of PCL/nylon-6 to prepare an optimized 3D multilayer composite scaffold, which can achieve sufficient mechanical and physical properties as well as cell viability and proliferation. Bone morphogenetic protein 12 (BMP-12) was added to promote the differentiation of MSCs, and mechanical stimulation was performed on the scaffold containing cells to imitate the natural function of tendons. The results showed that the synergistic effect of mechanical and biochemical stimulation was conducive to the enhancement of cell viability, proliferation, arrangement and tendon differentiation [104].

The sliding compression of the tendon is prone to rupture of the hydrogel, leading to changes in the polarity of macrophages related to biomaterials and inducing inflammation [105]. The hydrogel coating on the electrospun fiber membrane can enhance the support performance of the scaffold. Self-repairing and deformable HA hydrogel was attached to PCL electrospun nanofibers to form a composite double-layer patch. The electrospun nanofibers of the hydrogel could reduce inflammation and acted as a physical barrier to prevent adhesive tissue from invading the repaired tendon (Figure 3E) [106].

### 5.6. Electrospinning Combined with Other Advanced Technologies

The combination of electrospinning and other advanced technologies makes up for their respective technical defects and can give full play to the advantages of multitechnology cooperation. For example, electrospinning combined with 3D printing technology shows controlled mechanical properties and improves cell adhesion and proliferation [107]. Touré et al. innovatively electrospun fiber mats of PCL and polyglycerol sebacate (PGS) directly onto 3D-printed PCL–PGS blends containing bioactive glass on one side of the mesh. The scaffold showed good cell growth and penetration of 3D porosity, and excellent mechanical properties and degradation rates [108]. Guner et al. assembled fiber mats produced with rotary jet spinning and wet electrostatic spinning into a biphasic fiber carrier. The outer shell of the duplex support consisted of aligned PCL fibers to ensure better mechanical properties and strength. The support core consisted of PCL or PCL/gelatin fibers which were randomly aligned to provide a bionic structure suitable for cell adhesion. Subsequent in vitro studies have shown that the core fibers were randomly aligned in the aligned PCL fiber shell of the biphasic scaffold, which increased the initial adhesion, proliferation, and differentiation of the mouse fibroblast cell line. The alignment of the PCL fibers promoted elongation of the aligned fibers, thereby improving tendon tissue healing by directing cell proliferation and ECM deposition [109]. 

Multitechnology cooperation can improve performance in many aspects. Melt deposition modeling, melt electrospinning, and solution electrospinning can provide structural support, promote cell arrangement, and create bionic microenvironment to promote cell functioning. The surface modification of plasma can improve the hydrophilicity of the scaffold and creates a bionic microenvironment to enhance the ability of cell adhesion and proliferation [110]. Furthermore, Hakimi et al. applied both weaving and electrostatic spinning techniques to create strong implantable layered fibrous scaffolds for endogenous tendon repair, both consisting of the same fibrous material, polydioxanone (PDO). Due to the electrostatic spinning of the plates reinforced with woven layers, the scaffold could reach a maximum strength of 65 MPa and a maximum suture strength of 167 N, similar to that of a human rotator cuff tendon. Notably, the nanofibers of the composite scaffold mediated the biological activity and directed the behavior of the tendon cells in vitro and in vivo. Implantation into the rat model matched the mechanical properties of the tendon and progressively embedded it in dense tissue. The scaffold remained well integrated over time with no delamination (Figure 3F) [111]. This solved the problem of weak mechanical performance of electrospinning scaffold in tendon repair.

**Figure 3 polymers-15-01566-f003:**
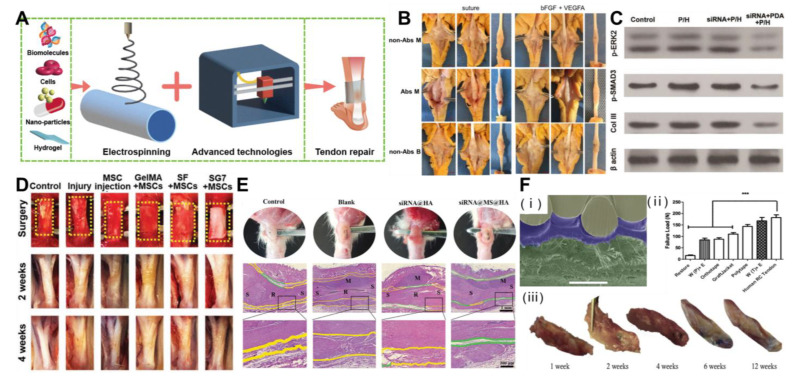
Treatment of tendon injury with composite electrospinning technology. (**A**) Scheme of bioactive materials and/or advanced technology combined with electrospinning for tendon repair. (**B**) Nanoparticle-coated suture improves the healing strength of injured tendon. Reprinted with permission from Ref. [92]. Copyright 2021 Elsevier Inc. (**C**) Significant inhibition of adhesion protein expression through PDA/ERK2-siRNA-mediated electrostatic spinning fibers. Reprinted with permission from Ref. [93]. Copyright 2018 Wiley-VCH GmbH. (**D**) Co-electrospinning cellulose and gelatin methacryloyl tablets with mesenchymal stem cells as seeds for tendon regeneration. The yellow dotted box represents the surgical site. Reprinted with permission from Ref. [36]. Copyright 2022 Wiley-VCH GmbH. (**E**) Hydrogel composite electrostatically spun nanofibers prevent tendon tissue adhesions and reduce inflammation. (green area: physical surrounding space; yellow area: adhesive tissue; S: su-ture site; R: repair site; M: anti-adhesive material). Reprinted with permission from Ref. [106]. Copyright 2022 Wiley-VCH GmbH. (**F**) Layered electrostatically spun braided surgical scaffolds for enhanced endogenous tendon repair. (*** *p* < 0.001). Reprinted with permission from Ref. [111]. Copyright 2021 IOP Publishing Ltd.

## 6. Advantages of Electrospinning 3D Structure in Tendon Repair at Different Stages

The 3D structure of electrospun fiber is similar to the natural ECM structure of a tendon, which can provide scaffold for tissue growth and support the organization and arrangement of cells in tissue. Meanwhile, it supports the infiltration of cells and nutrients, and promotes the proliferation and regeneration of tendon tissue. Through specific design, the electrospun 3D scaffold has its advantages in the three stages of inflammation, fibroblast proliferation, and remodeling (Figure 4A).

### 6.1. The Role of Electrospun 3D Structure in the Regulation of Inflammation

Inflammation coexists with the early degeneration of tendon, and the interference between immune cells and tendon fibroblasts will lead to poor tissue healing. The disordered fiber surface may affect the adhesion and gene expression of tendon cells, and make them produce pro-inflammatory signals [112]. In response to this problem, Schoenberger et al. used highly aligned polycaprolactone PCL scaffolds. Compared with randomly oriented nanofiber scaffolds, when primary human tendon fibroblasts were inoculated on PCL aligned nanofiber scaffolds, the expression of key transcription regulators and tendon ECM components collagen I, collagen III, anthocyanins, and disaccharides increased. At the same time, it reduced the expression of the ECM degradation matrix metalloproteinase and reduced the sensitivity of pro-inflammatory signals [113]. In addition, quantitative mechanical load response can promote the polarization of macrophages’ phenotype to an M2-like phenotype, thereby reducing the inflammatory reaction of tendons in vitro and in vivo. The ordered arrangement of PCL scaffolds better promotes tendon healing under mechanical loading (Figure 4B) [114]. In addition, Liu et al. proved that the expression level of the Cluster of Differentiation 68 (CD68) protein gradually decreased with the increase of the SF/PLLA mass ratio, the inflammatory reaction was also significantly inhibited (Figure 4C), and the collagen thickness gradually became thinner with the increase of the SF/PLLA mass ratio [58]. This provided strong support for the design of “immune regulation” characteristics of regenerative biomaterials. 

There is an inseparable relationship between the formation of adhesion and the expression of inflammation. It is an effective way to use anti-inflammatory materials to reduce the inflammatory reaction around the tendon. The nanofiber membrane with a polyethylene glycol/polycaprolactone/Ag NPs shell and hyaluronic acid/ibuprofen core prepared with coaxial electrospinning can reduce the adhesion of fibroblasts, and the core–shell structure can reduce pain and increase antibacterial activity by releasing ibuprofen and silver nanoparticles. At the same time, the combined effect of AgNPs and ibuprofen leads to the reduction of macrophages on the tendon tissue to reduce infection and inflammation, so as to meet the needs of an ideal antiadhesion barrier [101].

### 6.2. The Role of 3D Electrospun Structure in Tendon Proliferation

3D-oriented electrospun fibers exhibit flexible controllability in morphology and fiber density, have excellent biocompatibility in vitro, and are conducive to promoting cell adhesion, proliferation, and differentiation on its microstructure [115,116]. In the process of tendon proliferation, type I collagen (Col I) [117] and tendon regulatory protein (Tenomodulin, Tnmd) [118] are closely related to the tendon repair process and can be used to identify the degree of tendon repair. A bipolar metal flexible electrospun fiber membrane based on the membrane of metal-organic framework (MOF) is used for tendon–bone interface regeneration. Inspired by the biological gradient structure of the tendon–bone interface, the bipolar metal flexible electrospun fiber membrane of MOF is prepared using continuous electrospinning technology, which combines the regulation of osteoblast differentiation and angiogenesis characteristics to achieve synchronous regeneration. As a carrier, MOF can not only realize the sustainable release of metal ions, but also promotes osteogenesis and tendon formation on the scaffold. As a carrier, MOF can realize the sustainable release of metal ions, promote the upregulation of Col I (Figure 4Di) and Tnmd (Figure 4Dii) genes, and mark the proliferation and maturation of tendon cells. After 8 weeks of implantation, the morphology and inflammatory infiltration of the tissue at the interface of the flexible fibrous membrane were significantly improved (Figure 4Diii). This promoted the repair of tendon and bone tissue, as well as the reconstruction of fibrocartilage, and realized the simultaneous regeneration of multiple tissues at the damaged tendon–bone interface [119]. In addition, the pore size and porosity of the 3D electrospun scaffold determined the cell infiltration and regeneration ability. The high porosity of a 3D PCL scaffold can promote the rapid migration of MSCs into the scaffold, and further use of recombinant human connective tissue growth factor to stimulate MSCs can express higher levels of ligament/tendon gene type I collagen and tenomodulin, thus, promoting the proliferation and differentiation of mesenchymal stem cell ligament/tendon lineage [120]. 

### 6.3. The Role of 3D Structure of Electrospinning in Hemostasis

During the remodeling stage, the collagen synthesized from the scar at the tendon decreased, and the number of blood vessels and cells produced also decreased sharply. As type III collagen is not completely remolded into type I collagen, the fibrovascular scar produced by adult tendon in the initial healing stage will never be completely replaced [23]. The formation of scar will increase the risk of tendon rupture after repair [121]. In one study, the co-polymer (lactic acid/glycolic acid) electrospun fiber (Him-MFM) supported by the stearyl phosphate ethanolamine layer was connected with the CD11b^+^/CD68^+^ scar subpopulation membrane identified in the immune landscape after tendon injury to balance tissue damage. Him-MFM implanted into tendon can change the polarization of type I macrophages, reduce cell apoptosis and metabolic stress, and reduce inflammatory tendon cell response. It is worth noting that the hedging immune strategy increased the secretion of Interleukin-33 (IL-33) in the damaged tendon sheath by 4.36 times, activated the mucus-IL-33-Th2 axis, and directly provided regeneration niche for the proliferation of sheath stem cells [122]. In the mouse flexor tendon injury model, the tendon tissue of Him-MFM group was relatively smooth, no obvious fibrous tissue was found at the junction, and the degree of adhesion was low (Figure 4Ei). Previous studies have shown that the inhibition of the overexpression of myofibroblast markers α-SMA can prevent scar formation after tendon injury [123]. An immunofluorescence assessment of fibroblast infiltration showed that the number of α-SMA positive cells and normalized α-SMA intensity were significantly lower in the Him-MFM group than in the other groups, at 3.11 ± 1.97 and 14.01 ± 3.57, respectively, significantly reducing scar formation during tendon repair (Figure 4Eii), and comparable to mature tendons in terms of structure and function [122]. In addition, the electrospun polydioxanone support had high alignment or a random configuration. The healthy tendon fibroblasts cultured on the scaffold with highly arranged fibers for 7 days showed a unique elongated morphology. The scaffold simulating the alignment of collagen I fibrils with diameter in the process of tissue remodeling will not activate tendon-derived fibroblasts, thus, reducing fiber adhesion and scar formation [124].

**Figure 4 polymers-15-01566-f004:**
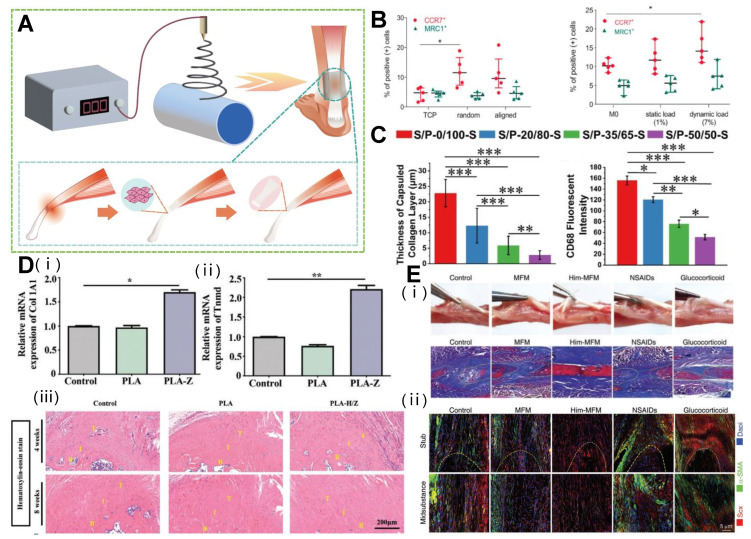
Electrospinning 3D structure used in different stages of tendon repair. (**A**) Application of electrospinning products in different stages of tendon repair. (**B**) Disorganized substrate topography and dynamic mechanical loading drive macrophage polarization toward a pro-inflammatory phenotype in vitro. (* *p* < 0.05). Reprinted with permission from Ref. [114]. Copyright 2020 Elsevier Inc. (**C**) Inflammatory response of different SF/PLLA nanoyarn-based woven fabrics 14 days after subcutaneous implantation. (* *p* < 0.05, ** *p* < 0.001, *** *p* < 0.0001). Reprinted with permission from Ref. [58]. Copyright 2022 Elsevier Inc. (**D**) MOF-based bipolar metal flexible electrospun fiber membrane increases tendon cell proliferation and maturation. (B: bone; I: interface; T: tendon; * *p* < 0.05, ** *p* < 0.01). Reprinted with permission from Ref. [119]. Copyright 2022 Wiley-VCH GmbH. (**E**) The Him-MFM promoted endogenous recovery of tendon and reduced adverse adhesion and scarring. Reprinted with permission from Ref. [122]. Copyright 2022 Wiley-VCH GmbH.

## 7. Patents Related to Electrospinning of Biodegradable Polymers

At present, many researchers are working on patented tendon repair technology using biodegradable polymers as materials. These techniques can increase the health of tendon tissue, which can improve joint mobility and function. We conducted a comprehensive search on Google patents using the keywords “biodegradable polymers” and “tendon repair” and yielded 155 results. In a careful review, 30 techniques related to tendon repair were listed, as shown in Table 1. Here, several biodegradable polymer products for tendon repair with different structures and different uses are presented. For example, Chinese Patent No. CN-103071185-A provides a bilayer-structured bionic-tendon-sheath membrane comprising a fibrous layer and a synovial layer, wherein the synovial layer of the bionic tendon sheath membrane contains HA or its metal salts, which enables the controlled release of HA or its metal salts, mimics the function of HA secretion by the inner layer of biological tendon sheaths, provides an environment with long-lasting lubricating effect for tendons, promotes endogenous healing of tendons, improves tendon healing quality, and improves postoperative tendon function. The fibrous layer mainly acts as a physical barrier to reduce peritendinous tissue adhesions and prevent exogenous healing. The biodegradable polymer material is also self-degrading and biocompatible, improving efficacy and reducing side effects [125]. US Patent No. US-2015190222-A1 comprises biodegradable/polymer fiber-based three-dimensional woven scaffolds with a woven structure comprising three regions, two ends designated for attachment to the structure, and an intermediate region used as a replacement ligament or tendon. The graft material can be surgically implanted into a patient’s ligament or tendon injury site to promote healing and repair of the injured ligament or tendon [126]. In addition, Canada Patent No. CA-2917427-C is a biodegradable polymer technology for tendon repair that addresses the integrity of the tendon by uniting biodegradable polymers through the suturing of the tendon tissue [127]. It is important to note that these patented technologies are only some of the current biodegradable polymer technologies that can be used for tendon repair, and many others are being researched and developed. In addition, different patented technologies may have different limitations and scope of application and should be selected and applied on a case-by-case basis.

## 8. Status and Commercialization of Clinical Trials of Biodegradable Polymer Electrospinning in Tendon Repair 

The effective application of biodegradable polymers in complex in vivo environments, and in particular the eventual clinical translation of the materials, still faces a number of key issues and challenges. Therefore, clinical trials of biodegradable polymers in tendon repair are summarized in Table 2. In order to adapt to medical applications in different in vivo environments, biodegradable polymers are made into different types of materials or products, such as micro- and nanoparticles, gels, and implantable devices, by means of self-assembly, chemical reactions, and molding processes. Methods of application of biodegradable polymers in tendon repair include use alone or in combination with other materials. For example, Park et al. randomized 40 patients to either a braided absorbable polylactide suture group or a braided non-absorbable polyethylene terephthalate suture group. After completing 12 months of follow-up, the absorbable sutures were found to be no worse than the non-absorbable sutures in repairing acute Achilles tendon ruptures [155]. PXL01 is a synthetic peptide derived serially from human lactoferrin with antibacterial and anti-inflammatory properties. Wiig et al. used a prospective, double-blind trial to verify that HA-loaded lactoferrin peptide (PXL01) improved hand recovery after flexor tendon repair [156]. Despite the potential advantages of biodegradable polymers in tendon repair, more research and experimental validation are needed before clinical application. Mentzel et al. [157] and Liew et al. [158] designed clinical protocols to validate the role of ADCON^®^-T/N in zone II flexor tendon repair, respectively, and did not obtain consistent results. Therefore, when performing tendon repair, the physician needs to consider the advantages and disadvantages of using biodegradable polymers to make the best treatment decision, depending on the specific condition and the patient’s circumstances.

Biological scaffolds for biomedical applications are not only available for research but also have real commercial potential. Products of animal origin or products prepared from natural biomaterials of animal origin as raw materials have been approved by the FDA 510(k) for the strengthening of soft tissues. Most commercial bio-scaffolds are prepared from natural biomaterials, such as collagen [167], or collagen cross-linked with synthetic biomaterials, such as polylactic acid [168], PLGA [169], and PLCL [170], which are synthetic polymers that have been approved by the Food and Drug Administration (FDA) for use in humans. The materials for commercially approved scaffolds are mainly type I collagen obtained from mammalian (bovine, porcine, equine) tissues, which have the natural structure and bioactivity to promote cell proliferation and tissue growth. Scaffolds that have been registered and used in the clinic for tendon reconstruction include TAPESTRY^®^ [171], GTR^®^ [172] and TenoMedTM [173]. The electrostatic spinning bio-scaffold has an ECM-like structure that facilitates cell adhesion and tissue regeneration, and offers structural advantages and biocompatibility in tendon repair. The electrostatic spinning bio-scaffold could be easily implanted into defective areas of injured tendon, for example, the TAPESTRY^®^ [171] Bio-integrative Implant developed by Embody, Inc. has been specifically designed for the management and protection of tendon injuries. TAPESTRY^®^ is designed for type I collagen and Poly (D, L-lactide) to provide a layer between the tendon and the surrounding tissue. Its constituent ingredients are currently approved by the FDA for clinical use. The medical collagen membrane with a bilayer structure developed by GTR Bio-Tech.Co., Ltd. is purified from bovine Achilles tendon tissue and has a bilayer structure, which matches the rate of degradation to the rate of tissue regeneration after implantation and is now in clinical use [172]. In addition, the composition and application of FDA registered biological scaffolds, as well as the manufacturer’s information, are summarized in Table 3. As the limitations of biomaterials in human applications are overcome, new commercial tendon repair products are being developed.

## 9. Conclusions

Electrospun nanofibers with controllable diameter, arrangement, high porosity, and a large surface area produced using electrospinning are usually easily able to meet the requirements of tendon repair tissue engineering, have the characteristics of low production costs and high efficiency, and are an ideal treatment choice for tendon injury [5,6,13,14,15]. Its mode of action involves providing a microenvironment for cell survival in vitro and expanding target cells, reducing the occurrence of inflammation at the injury site after implantation in vivo, and promoting the proliferation of tendon cells and the arrangement of collagen fibers.

A 3D electrospun-nanofiber scaffold based on the controllability of the electrospinning process can better simulate the structure and shape of ECM. A 3D electrospun-nanofiber scaffold with high porosity can be manufactured using direct electrospinning, post-processing technology, and by adjusting the fiber collection technology or combining these technologies [180]. In addition, researchers are exploring the use of new polymer materials, and mixing cells, bioactive molecules, and biomaterials into electrospun fibers to improve their performance and biological activity [181,182]. Natural biodegradable polymers play an integral role in tendon repair due to their tendon-tissue-mimicking structure and excellent biocompatibility. Currently, in the field of regenerative medicine, biological scaffolds with both biocompatibility and great mechanical properties have been prepared by combining natural biodegradable polymers with other polymers or with other biotechnologies such as 3D printing and freeze-drying.

The 3D structure of electrospun fiber provides several advantages for tendon repair, including the ability to provide a scaffold for tissue growth, support the infiltration of cells and nutrients, and be customized to match the characteristics of natural tissue. However, more research is needed to fully understand the potential mechanism of the positive effect of electrospun fibers on tissue repair. By optimizing its design and manufacturing methods, electrospun 3D scaffolds can display their capabilities in three stages of inflammation, proliferation, and remodeling after tendon injury. 

In the past, the use of 3D electrostatic spun wire scaffolds for tendon repair still posed many challenges. The weak mechanical properties of electrospun products prevented them from being used in hard tissue repair. The uncontrollable degradation rate of electrospun products made it a limitation in the application of regenerative medicine. However, as conventional electrospun technology is updated, there is increasingly great control over the diameter distribution of the membrane through operations such as the running track of the electrospun nozzle. At the same time, the solution concentration, electric/magnetic field, and conductivity of the solution are adjusted during the production process so that the properties of the fiber membrane are stable and controllable. In short, with the development and progress of electrospinning technology, the scalability and cost-effectiveness of electrospinning fibers used in tendon repair tissue engineering applications can be improved. 

## Figures and Tables

**Figure 1 polymers-15-01566-f001:**
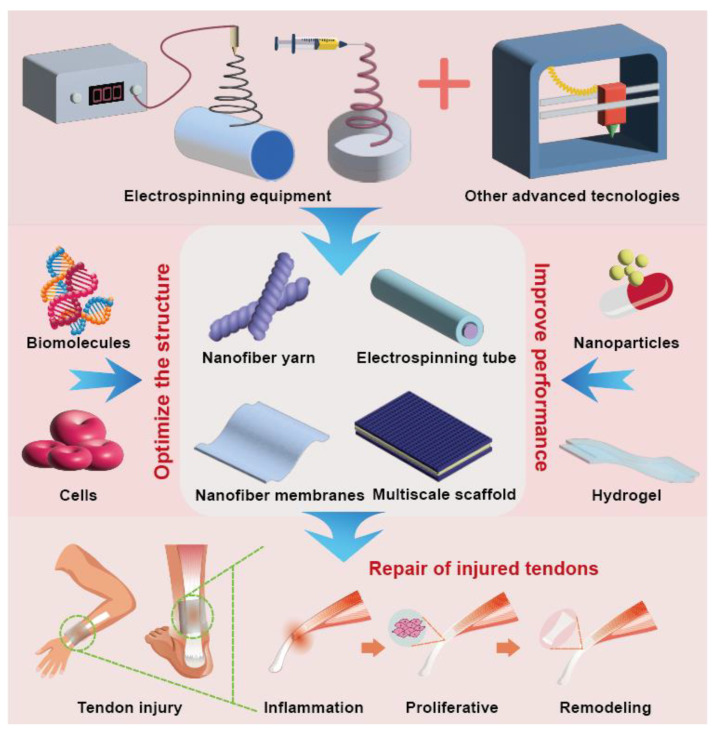
Diagram of electrospinning technology and its applications in the biomedical field.

**Table 1 polymers-15-01566-t001:** Patents for biodegradable polymers for tendon repair.

Patent Name	First Inventor	Publication	Patent ID	Ref.
Fiber-based scaffolds for tendon cell migration and regeneration	H. H. Lu	2021	WO-2021077042-A1	[128]
Tissue construct and method thereof	H. W. Ouyang	2006	WO-2006110110-A1	[129]
Medical implants and fibrosis-inducing agents	W. L. Hunter	2005	WO-2005065079-A2	[130]
Tissue repair scaffold	S. Downes	2017	US-9770529-B2	[131]
Biocompatible scaffold for ligament or tendon repair	F. Binnette	2014	US-8637066-B2	[132]
Tissue repair scaffold and device	S. Cartmell	2022	US-2022176014-A1	[133]
Biopolymer compositions, scaffolds, and devices	M. Francis	2020	US-2020338232-A1	[134]
Functional scaffold for tissue repair and regeneration	B. Han	2017	US-2017182080-A1	[135]
Ligament and tendon replacement constructs and methods for production and use thereof	C. T. Laurencin	2015	US-2015190222-A1	[126]
Synergetic functionalized spiral-in-tubular bone scaffolds	C. T. Laurencin	2010	US-2010310623-A1	[136]
Scaffold	O. Hakimi	2022	US-11517646-B2	[137]
Functionalized and crosslinked polymers	D. Gravett	2022	US-11440976-B2	[138]
Device for repair surgery of cylindrical organs, particularly ruptured tendons, comprising a therapeutic agent for stimulating regrowth, and method of producing such device	J. Buschmann	2020	US-10653819-B2	[139]
Biomimmetic nanofiber scaffold for soft tissue and soft tissue-to-bone repair, augmentation and replacement	H. H. Lu	2019	US-10265155-B2	[140]
A device for the repair of ruptured tendons comprising one or two needles and 60–80 cm of a supple thread in a resilient biodegradable polymer, preferably polylactic acid or polyglycolic acid	A. Leonard	2005	FR-2867380-A1	[141]
A device for repair surgery of cylindrical organs, particularly ruptured tendons	J. Buschmann	2018	EP-2747796-B1	[142]
Biological tissue connection and repair devices	A. Hoke	2020	US-9539009-B2	[143]
Promoter for regeneration of tendon-bone junction tissue or ligament-bone junction tissue	K. Tomita	2016	EP-2351574-B1	[144]
Fully synthetic implantable multiphased scaffold	H. H. Lu	2010	EP-2155112-A1	[145]
Tissue repair sleeve	N. Z. Zhang	2020	CN-210433516-U	[146]
A kind of tissue repair casing and its preparation method and application	N. Z. Zhang	2019	CN-109758197-A	[147]
Tissue-enhancement scaffold for use with soft tissue fixation repair systems and methods	M· Z· Morine	2022	CN-114601516-A	[148]
Tissue-reinforcing scaffold for use in soft tissue fixation repair	G. R. Whittaker	2020	CN-112107340-A	[149]
A kind of preparation method of multiple-dimensioned three-dimensional biological-tissue-engineering bracket	P. Zhao	2019	CN-106039417-B	[150]
For ligament or the method for tendon repair	B. Mathis	2017	CN-104254351-B	[151]
Simulated tendinous sheath film and preparation method thereof	C. Y. Fan	2013	CN-103071185-A	[125]
Tissue engineering scaffolds	A. S. Weiss	2021	CA-3165939-A1	[152]
Biopolymer scaffold implants and methods for their production	M. P. Francis	2019	CA-3079958-A1	[153]
Soft suture anchors	S. Rizk	2019	CA-2917427-C	[127]
Systems and methods for using structured tissue augmentation constructs in soft tissue fixation repair	T. Diab	2022	AU-2022202795-A1	[154]

**Table 2 polymers-15-01566-t002:** Clinical trials of biodegradable polymers in tendon repair.

Biodegradable Polymer Type	Study Type	Year	Cases (n)	Follow-Up	Clinical Manifestation	Ref.
Heterologous fibrin biopolymer (HFB)	Randomized Controlled	2022	84	21 days	HFB alone can effectively reduce the volume of edema and promote tendon repair.	[159]
Polylactic acid ester suture	Randomized Controlled	2022	37	12 months	Absorbable suture can be considered for repair of acute Achilles tendon rupture.	[155]
Bioamniotic membranes	Randomized Controlled	2019	89	12 months	Freeze-dried amniotic membrane transplantation was applied to promote healing of the flexor tendon in zone II and prevent adhesion.	[160]
Absorbable braided polyglactin suture	Randomized Controlled	2015	48	44 months	Absorbable suture repair is applicable to rupture of Achilles tendon.	[161]
PXL01 in sodium hyaluronate	Prospective, Randomised, Double-blinded	2014	138	12 months	Treatment with PXL01 in sodium hyaluronate improves hand recovery after flexor tendon repair surgery.	[156]
Rich fibrin matrix	Prospective, Randomized	2012	79	12 weeks	Platelet-rich fibrin matrix had no significant effect on tendon healing, tendon vessels, hand muscle strength or clinical evaluation scale.	[162]
HA	Randomized Controlled	2012	22	3 months	After repeated injection of HA for three months, the clinical outcome may be improved due to the reduction of adhesion in primary tendon repair.	[163]
Platelet-rich plasma fibrin matrix (PRPFM)	Randomized Controlled	2011	40	31 months	There was no significant difference in the repair effect of shoulder sleeve between PRPFM enhancement group and platelet-rich plasma (PRP) structure enhancement group.	[164]
ADCON-T/N	Randomised, Double-blinded	2001	59	6 months	Area II of ADCON-T/N treatment group is helpful for tendon repair and better interphalangeal movement of proximal end.	[158]
ADCON-T/N	Prospective randomized	2000	30	12 weeks	ADCON-T/N alone cannot improve postoperative adhesion of flexor tendon in zone II.	[157]
PLA	Randomized Controlled	1991	10	35 months	PLA promotes healing of tendon.	[165]
Absorbable polymer carbon fiber	Randomized Controlled	1989	71	4 years	Absorbable carbon fiber polymer can reduce infection, reduce the incidence of complications, and promote tendon healing.	[166]

**Table 3 polymers-15-01566-t003:** FDA-registered commercial products for tendon repair.

Product	Company	Application	Composition	Regulatory Approval	Company Website	Ref.
BioBlanket^®^	Kensey Nash, Inc.	BioBlanket^®^ is indicated for use duing rotator cuff repair surgery.	It is comprised of a single-layer porous, cross-linked collagen patch seperated from Bovine dermis.	FDA	https://www.tregistry.com, accessed on 3 March 2023.	[174]
BioBrace^®^	BIOREZ, Inc	BioBrace^®^ implant is an innovative bioinductive scaffold that is intended to reinforce soft tissue where weakness exists, and promote soft tissue healing	Consists of a type 1 collagen matrix and bioresorbable PLLA microfilaments.	FDA	https://biorez.com, accessed on 3 March 2023.	[175]
CuffPatch^®^	Arthrex, Inc.	CuffPatch^®^ is designed for rotator cuff repair, to support and accelerate healing of the tendons.	It is designed of Porcine small intestinal submucosa.	FDA	https://www.arthrex.com, accessed on 3 March 2023.	[176]
GTR^®^	GTR BioTech.Co., Ltd.	For guiding the regeneration of ruptured or damaged tendons.	It is made of collagen separated from bovine tendon tissue.	NMPA	https://www.gtrbio.cn, accessed on 3 March 2023.	[172]
OrthADAPT^™^	Pegasus Biologic, Inc	OrthADAPT^™^ Bioimplant is used for the repair, reinforcement and augmentation of soft tissues including tendons and ligaments.	It is a highly organized Type I collagen scaffold.	FDA	https://pegasusbio.com, accessed on 3 March 2023.	[177]
OrthoWrap^®^	MAST Biosurgery, Inc.	OrthoWrap^®^ Bioresorbable Sheet can be utilized for the management and protection of tendon injuries where there has been no substantial loss of tendon tissue.	It is a sheet made from an amorphous bioresorbable copolymer 70:30 Poly(L-lactide-co-D,L-lactide), commonly referred to as PLA.	FDA	https://mastbio.com, accessed on 3 March 2023.	[178]
TAPESTRY^®^	Embody, Inc.	TAPESTRY^®^ Biointegrated Implant is used as a protective layer between the tendon and surrounding tissues.	It is composed of collagen and poly(D,L-lactide).	FDA	https://embody-inc.com, accessed on 3 March 2023	[171]
TenoGlide^®^	Integra LifeSciences, Inc	TenoGlide^®^ Tendon protection sheet is used for tendon injury without substantial tissue loss, which can provide biocompatibility and sliding surface to protect tendon during healing.	It has cross-linked, highly purified type I collagen and glycosaminoglycan (GAG) porous matrix.	FDA	https://www.integranerve.com, accessed on 3 March 2023	[179]
TenoMed^™^	Exactech, Inc.	TenoMend^™^ Collagen Tendon Wrap is placed at the injured tendon to provide a protected environment and a sliding surface of the sheath for tendon healing.	It is an absorbable type I collagen matrix.	FDA	https://www.exac.com, accessed on 3 March 2023	[173]

## Data Availability

The data presented in this study are available on request from the corresponding author.

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
