# Peer review of "Biodegradable Polymer Electrospinning for Tendon Repairment"

_polymers, 2023, doi:10.3390/polym15061566_

Round 1
Reviewer 1 Report
Dear authors,
The submitted paper includes important and desglose information about the electrospinning technology for the treatment of tendon injuries. So, it is important to include the main developed concept in the title.
moreover,
-there are many typos to be corrected, in ex.: L105, L159-L166, L546-548.
-sentences to be clarify, in ex.: L135, L147.
-there are many acronyms to be defined: in ex.: DK, SLNM in L152 and perhaps some trade marks sings to be included.
-the used therm "cooperate" in L303 may not be the correct one in this case.
-table 1 use different size characters respect to the main text.
-the exhaustive use of other pubblications results (mainly due to the use of published specific data).
-and very important, the use of very extensive titles for the figures beside the agglomeration of information in figures. Please remember to take into account the instructions for authors "All Figures, Schemes and Tables should be inserted into the main text close to their first citation and must be numbered following their number of appearance (Figure 1, Scheme I, Figure 2, Scheme II, Table 1, etc.)" and "All Figures, Schemes and Tables should have a short explanatory title and caption".
Author Response
Thank you for your kind comment. Please see the reply letter from the attachment

Reviewer 2 Report
The review “Biodegradable Polymers used for the tendon repairment” covers an interesting and contemporary subject. The frequency of tendon injuries requires modern methods of recovery. The review presents the latest advances in electrostatic spinning technology for the treatment of tendon injuries. The article is well organized and can be accepted in its present form.
Just some typing mistakes should be corrected – lines 105, and 514. According to the instructions, a space is required before the square brackets of the references.
Author Response
Thanks for the reviewer’s approval and constructive suggestion. We have corrected all the typing mistakes in the manuscript. At the same time, the full text has been revised according to the format requirements of the magazine. A space has been added before the square brackets of the references.

Reviewer 3 Report
Biodegradable Polymers used for the tendon repairment is a very important topic and lies under the scope of this journal. However, some points must be considered before the final acceptance of the paper.
1. I request authors remove “the” from the title of the paper. It must be Biodegradable Polymers used for tendon repairment.
2. I request authors add a separate section to the paper focusing on the role of natural biodegradable polymers explored for tendon repairment.
3. I request authors add a section entitled “Limitations and future progress related to use of biodegradable polymers in tendon repairment” before the conclusion section of the paper.
4. I request authors add the current status of clinical trials related to the use of biodegradable polymers in tendon repairment.
5. I request authors add the current status of patents related to the use of biodegradable polymers in tendon repairment.
6. It is necessary to remove all the typographical and grammatical errors that are present in many sections of the manuscript before the final acceptance of the paper.
Thank you.
Author Response
Thanks for the reviewer’s approval and constructive suggestion. Please see the attachment.

Reviewer 4 Report
In this manuscript, Zhang et al. summarized the recent research progress of electrostatic spinning products in treating tendon injuries. They listed the 3D products with different structures and the composite scaffolds with bioactive materials manufactured by electrospinning used to treat tendon injuries, including clinical applications. The manuscript is worth publishing. However, there are some shortcomings, ambiguities, and questions that should be resolved:
(1) Authors mentioned the disadvantages of electrospun products in tendon repair in the abstract (line 22), but it was not discussed in the main text. They should take it as seriously as discussing the advantages of electrospun products to treat the disadvantages of electrospun materials in detail in one section. Giving readers an unbiased view of electrospun products in tendon repair is essential.
(2) Follow Comment 1. Some potential limitations involved in the manufacturing conditions were mentioned in the last part of the conclusions. Are these experimental parameters difficult to control? If yes, how to ensure the reproducibility of the experiment listed in the main text?
(3) The first three paragraphs in the conclusions are more like a summary of the electrospun products used for treating tendon injuries. It looks verbose to put in the conclusion section, and it would be better to move it elsewhere.
(4) The spelling of fibres and optimise in the abstract differs from fibers and optimize in the following parts. Please keep them consistent.
Author Response

(The authors gave the same response as above.)

Round 2
Reviewer 1 Report
The included modifications have significantly improved the text. However, the use of very extensive titles for the figures beside the agglomeration of information in figures, still do not take into account the instructions for authors, it is mandatory to use a short and explanatory title and caption. There is too much information in just one figure, I strongly suggest to disglose the figures in order to make more fluidic the lecture and enhanced the comprehension of the text.
In example, in Fig 1 "Diagram of electrospinning technology and its applications in the biomedical field" could be a more suitable title. Titles shoud only describe what the figure is or represents. A technical description can be included as long as it refers to symbols, nomenclatures, etc, that helps to "read" the figure (in ex.: round 1: dashed lines, round 2: solid lines")
While "Bio-degradable polymer electrospinning with different 3D structures is used for tendon repair. Bioactive molecules can optimize the structure of these products and improve their repair performance. The combination of other advanced technologies has greater potential in tendon repair." is an explanaition about the figure 1, this should be include in the main text.
After this last suggestion, the document can be accepted for publication.
Author Response
Please see the attachment.
Thank you very much for your email with the new comments. These suggestions are very helpful for us to improve the quality of the manuscript. We have revised the manuscript in accordance with the suggestions. For your rapid check, we list below the reply to the reviewers' comments point by point. And the important changes made in the revised manuscript have been highlighted in yellow.
